# Secondary Metabolites from *Aspergillus sparsus* NBERC_28952 and Their Herbicidal Activities

**DOI:** 10.3390/plants12010203

**Published:** 2023-01-03

**Authors:** Zhaoyuan Wu, Fang Liu, Shaoyong Ke, Zhigang Zhang, Hongtao Hu, Wei Fang, Shaoyujia Xiao, Yani Zhang, Yueying Wang, Kaimei Wang

**Affiliations:** 1Hubei Biopesticide Engineering Research Centre, Hubei Academy of Agricultural Sciences, Wuhan 430064, China; 2Department of Pharmacy, Medical College, Wuhan University of Science and Technology, Wuhan 430065, China

**Keywords:** *Aspergillus*, *Aspergillus sparsus* NBERC_28952, Aspersparins, sydonic acid, herbicidal activities, *Echinochloa crusgalli*, *Amaranthus retroflexus*

## Abstract

Fungi have been used in the production of a wide range of biologically active metabolites, including potent herbicides. In the search for pesticides of natural origin, *Aspergillus sparsus* NBERC_28952, a fungal strain with herbicidal activity, was obtained. Chemical study of secondary metabolites from NBERC_28952 resulted in the isolation of three new asperugin analogues, named Aspersparin A–C (**2**–**4**), and a new azaphilone derivative, named Aspersparin D (**5**), together with two known compounds, Asperugin B (**1**) and sydonic acid (**6**). The structures of these compounds were elucidated based on extensive spectroscopic data and single-crystal X-ray diffraction analysis. All of the isolated compounds were evaluated for their herbicidal activities on seedlings of *Echinochloa crusgalli* and *Amaranthus retroflexus* through Petri dish bioassays. Among them, compounds **5** and **6** exhibited moderate inhibitory activities against the growth of the roots and shoots of *E*. *crusgalli* seedlings in a dose-dependent manner, while **6** showed obvious inhibitory effect on seedlings of *A*. *retroflexus,* with an inhibitory rate of 78.34% at a concentration of 200 μg/mL. These herbicidal metabolites represent a new source of compounds to control weeds.

## 1. Introduction

Weeds can seriously threaten crop yields and cause huge economic losses [1]. A lot of measures, including cultivation, mechanical operations, and the application of chemicals, have been implemented to control weeds. However, the excessive use of chemical herbicides leads to residue seepage into the environment and the increased resistance of weeds to such compounds [2]. Given the present concerns about the negative impacts of chemical herbicides on human health and the environment, it is necessary to develop safer compounds to ensure the sustainability of crop production.

Natural products have played an important role in the development of pesticides for crop protection [3]. An analysis showed that 41.8% of the pesticides listed in the registry of the Environmental Protection Agency were based on active ingredients developed with natural products [4]. Natural products might be a source of new herbicidal entities with potentially new modes of actions [5]. Fungi are considered as one of the richest sources of natural products among living organisms [6]. The phytotoxins produced by fungi are often suitable for the pathogenesis or infection of weeds [7]. Fungi of *Aspergillus* spp. have been shown to be excellent sources of new natural chemicals [8,9,10,11,12], some of which have shown promising herbicidal activities. For example, Asperalacid D, a new natural sesquiterpenoid from *Aspergillus alabamensis*, showed higher plant growth inhibitory activity on wheat root and shoot elongation than terbutryn [13]. Additionally, 8-methoxycichorine, 8-epi-methoxycichorine, and N-(4′-carboxybutyl) cichorine, three novel cichorine analogues with an isoindolinone skeleton, obtained from *A*. *nidulans*, exhibited superior phytotoxicity to cichorine on the leaves of *Zea mays* and *Medicago polymorpha* [14]. Dihydrosterigmatocystin, isolated from an alga-derived fungus, *Aspergillus versicolor*, caused leaf necrosis and plant wilting in *Amaranthus retroflexus*, with a MIC of 24.5 μM, i.e., almost four-fold stronger than that of glyphosate [15]. This indicates the potential for discovering novel herbicidal compounds from the secondary metabolites of fungi in the genus *Aspergillus*. During the course of screening for microbial secondary metabolites possessing bioactivities, we isolated some active compounds with new structures [16,17,18]. In an ongoing effort to discover bioactive natural products obtained from microbes, our attention was drawn to the fungus *A*. *sparsus* NBERC_28952 because of its potent herbicidal activity. In this study, we describe the isolation, structure, and herbicidal activities of the compounds obtained from NBERC_28952. To the best of our knowledge, this is the first report on this topic.

## 2. Results

### 2.1. Compounds Identification

HPLC-MS- and UV-guided isolation of the EtOAc extract of *A*. *sparsus* revealed the presence of four new compounds, Aspersparins A−D (**2**−**5**), together with two known ones, Asperugin B (**1**) [19] and sydonic acid (**6**) [20] (Figure 1).

Compounds **1**−**4** were analogues, all of which were obtained as pale yellow amorphous solids. Using HR-ESI-MS (*m*/*z* 387.2170, (M+H)^+^) analysis, the molecular formula of compound **1** was determined to be C_23_H_30_O_5_ (see Appendix A). The ^1^H-, ^13^C-NMR and HSQC spectra (Table 1 and Table 2, Appendix A) of **1** showed signals for a pentasubstituted benzene (δ_H_ 7.17 (1H, s, H-5); δ_C_ 159.7 s, 157.9 s, 137.8 s, 134.6 s, 117.6 d, 113.6 s), two aldehydes (δ_H_ 10.73 (1H, s, H-7), δ_H_ 10.11 (1H, s, H-8); δ_C_ 196.5 d, 193.1 d), together with a farnesyl side chain containing four methyls (δ_C_ 26.0, 17.9, 16.5, 16.2), five methylenes (δ_C_ 69.5, 40.5, 40.4, 27.5, 27.1), three ethylenic methines (δ_C_ 125.2, 124.7, 120.8), and three ethylenic quaternary carbons (δ_C_ 143.4, 136.0, 131.8). The structure of the farnesyl side chain was further deduced from ^1^H-^1^H COSY (see Figure 2 and Appendix A) and HMBC (see Figure 2 and Appendix A) correlations. HMBC couplings of the methylene protons at δ_H_ 4.78 (2H, d, *J* = 7.3 Hz, H_2_-1′) to a quaternary carbon at δ_C_ 137.8 (C-1) indicated *O*-prenylation of the aromatic system. ^1^H-^1^H COSY correlations of H_2_-1′/H-2′, H_2_-4′/H_2_-5′/H-6′, and H_2_-8′/H_2_-9′/H-10′ suggested the presence of three olefinic bonds at C-2′/C-3′, C-6′/C-7′, and C-10′/C-11′, respectively. HMBC correlations of H_2_-1′ to C-1, C-2′ (δ_C_ 120.8, d) and C-3′ (δ_C_ 143.4, s), H-2′ (δ_H_ 5.52, t, *J* = 7.3 Hz) to C-4′ (δ_C_ 40.5, t) and Me-15′ (δ_C_ 16.5, q), H_2_-4′ (δ_H_ 2.02, m) to C-3′, C-5′ (δ_C_ 27.5, t), C-6′ (δ_C_ 124.7, d) and Me-15′, H-6′ (δ_H_ 5.08, m) to C-7′ (δ_C_ 136.0, s), C-8′ (δ_C_ 40.4, t), H_3_-14′ (δ_H_ 1.58, s) to C-6′, H_2_-8′ (δ_H_ 1.95, t, *J* = 7.2 Hz) to C-7′, C-10′ (δ_C_ 125.2, d) and Me-14′ (δ_C_ 16.2, q), H_2_-9′ (δ_H_ 2.05, overlapped) to C-8′, C-10′ and C-11′ (δ_C_ 131.8, s), H-10′ (δ_H_ 5.08, m) to Me-12′ (δ_C_ 26.0, q) and Me-13′ (δ_C_ 17.9, q) confirmed the structure of a farnesyl side chain. As required by the molecular formula, two hydroxyl groups were located at the phenyl ring. Moreover, HMBC couplings of H-5 to C-1 (δ_C_ 137.8, s), C-3 (δ_C_ 113.6, s), C-6 (δ_C_ 157.9, s) and C-8 (δ_C_ 193.1, d), H-7 to C-2 (δ_C_ 159.7, s) and C-3, and H-8 to C-3, C-4 (δ_C_ 134.6, s) and C-5 (δ_C_ 117.6, d) revealed the substitution positions. NOE correlations (see Figure 3 and Appendix A) from H_2_-4′ to H-2′ and from H_2_-8′ to H-6′ supported the existence of *E*-configurations of the two double bonds of C-2′/C-3′ and C-6′/C-7′. Thus, the structure of **1** was established, and the compound was named Asperugin B [19]. The ^1^H- and ^13^C-NMR spectra data are presented in detail, for the first time, in this paper.

Compound **2** had a molecular formula of C_23_H_30_O_5_, as inferred by its HR-ESI-MS (*m*/*z* 387.2163, (M + H)^+^) (Appendix A), sharing the same molecular formula with compound **1**. The ^1^H- and ^13^C-NMR spectra (Table 1 and Table 2, Appendix A) of the phenolic part of compound **2** were almost the same as those of **1**. However, in the 1D-NMR and HSQC (see Appendix A) spectra of **2**, the farnesyl side chain included four methyls (δ_C_ 23.0, 23.0, 16.5, 16.3), four methylenes (δ_C_ 69.4, 39.9, 39.8, 26.5), five methines (δ_C_ 140.1, 126.2, 124.8, 120.8, 32.1), and two quaternary carbons (δ_C_ 143.6, 136.6), in contrast to compound **1**. The ^1^H-^1^H COSY cross-peaks (Figure 2 and Appendix A) of H_2_-1′/H-2′, H_2_-4′/H_2_-5′/H_2_-6′, and H-8′/H-9′/H-10′/H-11′/Me-12′ (Me-13′) suggested the presence of three olefinic bonds at C-2′/C-3′, C-7′/C-8′, and C-9′/C-10′, respectively. HMBC couplings (Figure 2 and Appendix A) of H_2_-1′ (δ_H_ 4.80, *J* = 7.3 Hz) to C-2′ (δ_C_ 120.8, d) and C-3′ (δ_C_ 143.6, s), H-2′ (δ_H_ 5.50, td, *J* = 7.3, 1.2) to Me-15′ (δ_C_ 16.3, q), H_2_-4′ (δ_H_ 1.97, t, 7.4) to C-2′, C-3′ and Me-15′, H_2_-5′ (δ_H_ 1.47, m) to C-4′ (δ_C_ 39.9, t) and C-6′ (δ_C_ 39.8, t), H_2_-6′ (δ_H_ 1.88, t, 7.5) to C-4′ and Me-14′ (δ_C_ 16.5, q), H-8′ (δ_H_ 5.71, d, *J* = 10.8 Hz) to C-6′ and Me-14′, H-10′ (δ_H_ 5.53, dd, *J* = 15.2, 7.1 Hz) to C-8′ (δ_C_ 126.2, d), H_3_-12′/13′ (δ_H_ 0.99, d, *J* = 6.8 Hz) to C-10′ (δ_C_ 140.1, d) and C-11′ (δ_C_ 32.1, d) established the structure of the farnesyl side chain as shown in Figure 1. Moreover, HMBC correlations of 2-OH (δ_H_ 12.84, s) to C-1 (δ_C_ 137.6, s), C-2 (δ_C_ 159.5, s), and C-3 (δ_C_ 113.6, s) further confirmed the substitution positions of the phenyl ring. NOEs (see Figure 3 and Appendix A) from H_2_-4′ to H-2′, from H-8′ to H_2_-6′, and from H-9′ (δ_H_ 6.21, ddd, *J* = 15.1, 10.8, 1.0 Hz) to H-11′ (δ_H_ 2.33, sext, *J* = 6.8 Hz) supported the existence of *E*-configurations of all the double bonds. The structure of **2** was thus determined, and the compound was named Aspersparin A.

The molecular formula of compound **3** was also determined to be C_23_H_30_O_5_ based on its HR-ESI-MS (*m*/*z* 387.2167, (M + H)^+^) (see Appendix A), which indicated that it was an isomer of **1** and **2**. However, the 1D-NMR spectra (see Table 1 and Table 2, Appendix A) and HSQC spectrum (Appendix A) showed some differences in the farnesyl side chain. ^1^H-^1^H COSY cross-peaks (see Figure 2 and Appendix A) of H_2_-1′/H_2_-2′, H-4′/H_2_-5′/H_2_-6′, and H-8′/H-9′/H-10′/H-11′/Me-12′ (Me-13′) suggested the presence of three olefinic bonds at C-3′/C-4′, C-7′/C-8′, and C-9′/C-10′, respectively. Supported by HMBC correlations (see Figure 2 and Appendix A) from H_2_-2′ (δ_H_ 2.46, t, *J* = 7.2 Hz) to C-1′ (δ_C_ 72.2, t), C-3′ (δ_C_ 132.3, s), C-4′ (δ_C_ 127.5, d) and Me-15′ (δ_C_ 16.3, q), H-4′ (δ_H_ 5.24, td, *J* = 6.9, 1.1 Hz) to C-6′ (δ_C_ 40.7 t) and Me-15, H_2_-6′ (δ_H_ 2.04, overlapped) to C-5′ (δ_C_ 27.3, t), C-7′ (δ_C_ 136.4, s), C-8′ (δ_C_ 126.2, d) and Me-14′ (δ_C_ 16.6, q), H-8′ (δ_H_ 5.77, d, *J* = 10.8 Hz) to C-6′ and Me-14′, H-9′ (δ_H_ 6.21, ddd, *J* = 15.1, 10.8, 1.0 Hz) to C-11′ (δ_C_ 32.1, d), and H-10′ (δ_H_ 5.50, dd, *J* = 15.1, 7.0 Hz) to C-8′ and Me-12′/Me-13′ (δ_C_ 23.0, q), the core structure of **3** was established. NOEs (Figure 3 and Appendix A) from H-4′ to H_2_-2′ and from H-8′ to H_2_-6′ supported the presence of *E*-configurations for the double bonds at C-3′/C-4′ and C-7′/C-8′. The *E*-configuration for C-9′/C-10′ was verified by the coupling constant (*J* = 15.1 Hz) of H-9′ and H-10′. The structure of **3** was thus determined, and the compound was named Aspersparin B.

Compound **4** had a molecular formula of C_25_H_34_O_6_ according to HR-ESI-MS (*m*/*z* 453.2255, (M + Na)^+^) (Appendix A). NMR signals (see Table 1 and Table 2, Appendix A) of δ_H_ at 5.32 (s, H_2_-8) and 2.86 (s, H_3_-2″), δ_C_ at 63.2 (t, C-8), 170.6 (s, C-1″) and 20.9 (q, C-2″), together with HMBC correlations (see Figure 2 and Appendix A) of H_2_-8 to C-3 (δ_C_ 113.4, s), C-4 (δ_C_ 137.0, s), C-5 (δ_C_ 111.3, d), C-1″, and H_3_-2″ to C-1″ revealed the presence of a –CH_2_OCOCH_3_ at C-4. The farnesyl side chain of compound **4** was assumed to be identical with that of compound **2**; this hypothesis was supported by almost identical ^1^H-, ^13^C-NMR chemical shifts and NOE correlations of this compound (Appendix A). Thus, the structure of **4** was determined, and the compound was named as Aspersparin C.

Compound **5** was isolated as yellow crystalline powder. Its molecular formula was determined to be C_22_H_26_O_5_ by HR-ESI-MS (*m*/*z* 371.1856, (M + H)^+^) (Appendix A), which implied ten degrees of unsaturations. The ^1^H-NMR spectrum (Table 3, Appendix A) indicated the presence of four methyl groups (δ_H_ 2.28 (3H, s, H_3_-10), 1.69 (3H, s, H_3_-9), 1.16 (3H, d, *J* = 6.6 Hz, H_3_-9′), and 0.86 (3H, t, *J* = 7.3 Hz, H_3_-8′)), five *sp*^3^ methylene protons (δ_H_ 1.20~1.56 (10H, m, H_2_-3′, 4′, 5′, 6′, 7′)), three uncoupled olefinic protons (δ_H_ 8.72 (1H, s, H-1), 6.44 (1H, s, H-4), and 5.35 (1H, s, H-5)), and one *sp*^3^ methine proton (δ_H_ 3.58 (1H, m, H-2′)). The ^13^C-NMR (see Table 3, Appendix A) and HSQC (see Appendix A) spectra displayed the signals of four methyl carbons (δ_C_ 24.6, 17.9, 13.8 and 13.0), five aliphatic methylene carbons (δ_C_ 33.3, 28.9, 27.5, 26.3 and 22.1), one aliphatic methine carbon (δ_C_ 43.3), three olefinic methine carbons (δ_C_ 153.2, 108.1 and 103.7), one oxygenated quaternary carbon (δ_C_ 87.7), five olefinic quaternary carbons (δ_C_ 165.2, 160.3, 146.0, 124.2 and 111.1), one ester carbonyl carbon (δ_C_ 168.3), and two *α*,*β*-unsaturated ketone carbonyl carbons (δ_C_ 200.7 and 191.5). Three carbonyls and eight olefinic carbons accounted for seven degrees of unsaturation. The remaining three degrees of unsaturation supported the existence of three rings. HMBC correlations (Appendix A) of H-1 to C-3 (δ_C_ 160.3), 4a (δ_C_ 146.0), 8a (δ_C_ 111.1), H-4 to C-3, 5 (δ_C_ 103.7), 8a, and H-5 to C-4 (δ_C_ 108.1), C-7 (δ_C_ 87.7), and C-8a suggested the presence of an azaphilone skeleton. Moreover, the presence of a ketone aliphatic side chain was supported by the HSQC, COSY (Appendix A) and HMBC spectra. A comparison of the ^1^H- and ^13^C-NMR data of **5** with those of didehydrochermesinone B [21] showed that the former contained four more CH_2_ in the fatty chain. Accordingly, the planar structure of **5** was established. The absolute configuration of **5** was determined by single-crystal X-ray diffraction analysis. An Olex2 plot is shown in Figure 4, illustrating the absolute configuration of **5** to be 7R, 2′S with Flack parameter –0.07 (10) (CCDC 2219385). The structure of **5** was finally determined and the compound was named Aspersparin D.

### 2.2. Evaluation of Herbicidal Activities

The herbicidal activities of the isolated compounds toward seedlings of *E*. *crusgalli* and *A*. *retroflexus* were assessed at 200 μg/mL using 2,4-dichlorophenoxy acetic acid (2,4-D) as the positive control through Petri dish bioassays (Table 4). Of the tested compounds, Aspersparin D (**5**) and sydonic acid (**6**) exhibited moderate inhibitory activities against the growth of the roots and shoots of *E*. *crusgalli* seedlings (with inhibitory rates ranging from 40% to 60%). Compound **6** showed a lower inhibition rate to *E*. *crusgalli* seedlings than **5**, but a much higher one to the radicle and germ of *A*. *retroflexus* (78.34%), similar to that of 2,4-D (80.70%).

Compounds **5** and **6** exhibited much stronger inhibitory activities against the growth of *E*. *crusgalli* and *A*. *retroflexus* seedlings than the other tested compounds. Therefore, further investigation of herbicidal activities of **5** and **6** was conducted. The inhibitory effects of gradient concentrations (12.5, 25, 50, 100, 200 μg/mL) of these compounds on the growth of seedlings of the two weed varieties were assayed. As shown in Figure 5, compounds **5** and **6** inhibited the growth of the seedlings of the two weeds in a dose-dependent manner. Furthermore, in the concentration range of 50–200 μg/mL, **5** and **6** displayed moderate inhibitory activities against the growth of shoots of *E*. *crusgalli* seedlings. In addition, in contrast to the positive control 2,4-D, the inhibitory effects of **5** and **6** on shoots of *E*. *crusgalli* seedlings were stronger than on the roots. Both **5** and **6** exhibited good inhibition on *A*. *retroflexus* seedlings, i.e., higher than 50% inhibition at a concentration of 50 μg/mL. Interestingly, when the concentration was below 50 μg/mL, the inhibitory effect on *A*. *retroflexus* of **5** was much lower than that of **6**, but it increased more rapidly with an increase of concentration, achieving over 60% inhibition at 100 μg/mL and over 75% inhibition at 200 μg/mL.

## 3. Discussion

Compounds **1**–**4** were asperugin analogues which showed weak inhibitory activities against the growth of *E*. *crusgalli* and *A*. *retroflexus* seedlings at the concentration of 200 μg/mL. Compounds **1**–**3** were double-bond isomers with the same structure in the phenyl part, while compound **4** had a –CH_2_OCOCH_3_ located at the phenyl ring. The bioassay results showed that **4** exhibited higher inhibition on the tested weeds than compounds **1**–**3**. Therefore, the substituents on the benzene ring might play an important role in the herbicidal activities of asperugin analogues. This is the first report on the herbicidal activities of asperugin analogues. These compounds may prove to be promising in the research and development of natural or plant-derived herbicides.

Azaphilones are a large family of metabolites derived from fungi with variable structures, exhibiting a wide range of biological activities [22]; however, at present, there are relatively few reports about their herbicidal activities [23]. As reported in our previous study [24], Chaetomugilin O, an azaphilone with tetrahydrofuranone, exhibited the most potent inhibition among the tested azaphilones of seedling growth of several weeds. In our study, Asperaprin D (**5**) was identified as a new azaphilone derivative with a tetrahydrofuranone group. The compound showed moderate inhibitory activity against *E*. *crusgalli* and *A*. *retroflexus* seedlings. Based on a comparison of the structures of Chaetomugilin O and Asperaprin D (**5**), we speculate that the presence of a tetrahydrofuranone group may be an important factor for the growth-suppression activity of azaphilones, with the side chains connected to C-3 and C-7 also influencing the inhibitory effects. In any case, further studies on the structure–activity relationship and structural modifications are needed to verify this hypothesis. Previous studies have examined the antibacterial activity [25] and cytotoxic activity [26] of sydonic acid (**6**), a bisabolane-type sesquiterpene produced by several fungi. In the present study, sydonic acid (**6**) was found to have herbicidal activities, showing moderate inhibitory effects on *E*. *crusgalli* seedlings and obvious inhibitory effects on *A*. *retroflexus* seedlings. This is the first time that compounds **5** and **6** have been reported as herbicidal fungal metabolites. These findings could expand the application of these compounds in the agricultural domain and broaden our knowledge on the possible structures of natural herbicides. However, whether the compounds are useful as mycoherbicides remains to be determined by further study via pot experiments in a greenhouse and field experiments.

## 4. Materials and Methods

### 4.1. General Experimental Procedures

NMR spectra, including HSQC, HMBC, COSY, and NOESY, were recorded on a Bruker AVANCE-500 instrument with tetramethylsilane (TMS) as an internal standard (Bruker BioSpin group, Rheinstetten, Germany). ESI-MS and HR-ESI-MS data were obtained on a Waters LC-MS (Waters Corporation, Milford, MA, USA) and Thermo Q-T of Micromass (Thermo Electron Corporation, Waltham, MA, USA) spectrometers, respectively. Preparative HPLC was carried on a Waters 2767 Autopurification System (Waters Corporation, Milford, MA, USA) coupled with a DAD detector, using a Sunfire Prep C18 OBD (5 μm, 19 × 250 mm, Waters Corporation, Milford, MA, USA) column. X-ray crystallography analysis was conducted on a Bruker D8 VENTURE (Bruker Corporation, Billerica, MA, USA) single-crystal diffractometer.

### 4.2. Fungal Material

*Aspergillus sparsus* strain NBERC_28952 was isolated from a soil sample collected from Xihui Park in Wuxi City, Jiangsu Province, China, in March 2012. The strain was identified 16S rRNA sequence analysis. A voucher strain was preserved at Hubei Biopesticide Engineering Research Center, Hubei Academy of Agricultural Sciences, Wuhan, China.

### 4.3. Cultivation, Extraction, and Isolation

A stock of *A*. *sparsus* strain NBERC_28952, previously stored at −86 °C, was streaked on a potato-dextrose agar plate and incubated at 25 °C until good growth was observed. Culture discs with an internal diameter of 5 were made with a sterile stainless-steel puncher mm. Five culture discs were inoculated into separate 500 mL flasks containing 100 mL of seed medium each. The seed culture was shaken at 120 rpm at 28 °C. After 96 h of cultivation, seed cultures (10%) were transferred to 500 mL Erlenmeyer flasks containing 100 mL medium with the following composition: 6.25 g/L malt extract, 6.25 g/L maltose, 1.0 g/L yeast extract, 0.625 g/L soybean peptone, 1.25 g/L KH_2_PO_4_, and 1.25 g/L MgSO_4_. The inoculated flasks were incubated at 28 °C for 96 h on a rotary shaker (120 rpm).

The fermented material (10.0 L) was extracted using ethyl acetate (3 × 10.0 L) following three 30 min periods of stirring [16]. The organic solvent was filtrated and then concentrated in vacuo to obtain a crude extract (3.6 g).

The ethyl acetate extract was dissolved in acetonitrile and subjected to silica gel chromatography through progressive column elution with petroleum ether (PE−EtOAc (5:1–1:1, *v*/*v*)) to obtain five major fractions (Fr.1–Fr.5). Fr.1 (PE−EtOAc, 5:1, *v*/*v*, 1.1 g) was then dissolved in dichloromethane and subjected to silica gel CC elution with petroleum ether (PE−EtOAc (50:1–2:1, *v*/*v*)) to obtain four subfractions (Fr.1.1–Fr.1.4). Subfraction Fr.1.3 was profiled with reversed-phased HPLC (Sunfire^®^, Prep C18 OBD, 19 × 250 mm, 5 µm, 27 mL/min) using a gradient solvent system from 40–100% CH_3_CN for 30 min, yielding compounds **1** (4.98 mg), **2** (21.97 mg), and **3** (13.66 mg). Compound **4** (5.23 mg) was purified from subfraction Fr.1.4 by preparative HPLC from 40% to 100% CH_3_CN for 30 min. Compounds **5** (15.86 mg) and **6** (22.92 mg) were separated from Fr. 4 (PE−EtOAc, 2:1, *v*/*v*, 0.6 g) by preparative HPLC from 30 to 100% CH_3_CN for 30 min.

Asperugin B (**1**): Faint yellow amorphous solid; ^1^H-NMR (500 MHz, CO(CD_3_)_2_) and ^13^C-NMR (125 MHz, CO(CD_3_)_2_). See Table 1 and Table 2. HR-ESI-MS (positive mode): *m*/*z* 387.2170, [M + H]^+^ (Calcd for C_23_H_31_O_5_, 387.2166).

Aspersparin A (**2**): Faint yellow amorphous solid; ^1^H-NMR (500 MHz, CO(CD_3_)_2_) and ^13^C-NMR (125 MHz, CO(CD_3_)_2_). See Table 1 and Table 2. HR-ESI-MS (positive mode): *m*/*z* 387.2163, [M + H]^+^ (Calcd for C_23_H_31_O_5_, 387.2166).

Aspersparin B (**3**): Faint yellow amorphous solid; ^1^H-NMR (500 MHz, CO(CD_3_)_2_) and ^13^C-NMR (125 MHz, CO(CD_3_)_2_). See Table 1 and Table 2. HR-ESI-MS (positive mode): *m*/*z* 387.2167, [M + H]^+^ (Calcd for C_23_H_31_O_5_, 387.2166).

Aspersparin C (**4**): Faint yellow amorphous solid; ^1^H-NMR (500 MHz, CO(CD_3_)_2_) and ^13^C-NMR (125 MHz, CO(CD_3_)_2_). See Table 1 and Table 2. HR-ESI-MS (positive mode): *m*/*z* 453.2255, [M + Na]^+^ (Calcd for C_25_H_34_NaO_6_, 453. 2248).

Aspersparin D (**5**): yellow crystalline powder; [α]${}_{D}^{22}$ + 0.48 (c 0.20, MeOH); ^1^H-NMR (500 MHz, CD_3_OD) and ^13^C-NMR (125 MHz, CD_3_OD). See Table 3. HR-ESI-MS (positive mode): *m*/*z* 371.1857, [M + H]^+^ (Calcd for C_22_H_27_O_5_, 371.1853).

Sydonic acid (**6**): white powder; ^1^H-NMR (500 MHz, CD_3_OD): 0.84 (6H, d, *J* = 6.6 Hz), 1.16 (2H, m), 1.22 (1H, m), 1.38 (1H, m), 1.54 (1H, m), 1.63 (3H, s), 1.81 (ddd, *J* = 4.5, 11.4, 13.5 Hz), 1.97 (1H, dt, *J* = 4.5, 13.5 Hz), 7.28 (1H, d, *J* = 8.1 Hz), 7.41 (1H, d, *J* = 1.6 Hz), 7.48 (1H, dd, *J* = 1.6, 8.1 Hz). ^13^C-NMR (125 MHz, CD_3_OD): 21.5 (t), 21.6 (q), 21.7 (q), 27.5 (d), 27.6 (q), 39.1 (t), 42.3 (t), 76.6 (s), 117.3 (d), 121.2 (d), 126.4 (d), 130.3 (s), 136.6 (s), 155.6 (s), 168.6 (s). The ^1^H- and ^13^C-NMR data were in good accordance with the reported data [20].

### 4.4. X-ray Crystallographic Data Analysis

Crystals of compound **5** were obtained from methanol. A suitable crystal was selected and examined on a Bruker D8 VENTURE single-crystal diffractometer. The crystal was kept at 100.0 K during data collection. Using Olex2 [27], the structure was determined using the XM [28] structure solution program with Dual Space and refined with the XL [28] refinement package using Least Squares minimization. Crystallographic data for **5** were deposited in the Cambridge Crystallographic Data Centre with the deposition number CCDC 2219385.

Crystal data of compound **5**: C_22_H_26_O_5_ (M = 370.43 g/mol); orthorhombic, space group P212121 (no. 19), a = 5.3508 (3) Å, b = 9.2068 (6) Å, c = 38.888 (2) Å, V = 1915.8 (2) Å3, Z = 4, T = 100.0 (1) K, μ (Synchrotron) = 0.467 mm^−1^, Dcalc = 1.284 g/cm^3^, 11,234 reflections measured (7.912° ≤ 2θ ≤ 124.428°), 4420 unique (Rint = 0.0409, Rsigma = 0.0422), which were used in all calculations. The final R1 was 0.0314 (I > 2σ (I)) and wR2 was 0.0866 (all data). Flack parameter: −0.07 (10).

### 4.5. Herbicidal Activity Assays

The isolated compounds were screened for potential herbicidal activity against a monocotyledonous weed, *E*. *crusgalli*, and a dicotyledonous weed, *A*. *retroflexus*, using a Perish dish bioassay [29,30]. Distilled water was used as a negative control, while 2,4-D was used as a positive control. The weed seeds were placed at 28 °C for 24~48 h for germination. Then, germinated seeds with consistent status were selected for use in subsequent tests.

*Experiment 1*. Compounds **1**–**6** and 2,4-D were dissolved in acetone at concentrations of 200 μg/mL. The acetone solutions (3 mL) were transferred into separate sterilized 90 mm Petri dishes with filter paper, while 3 mL acetone was used for the negative control. After complete evaporation of the acetone, distilled water (3 mL) was added to each petri dish. Next, 15 seeds were evenly placed on the filter paper. The Petri dishes were covered with lids and placed in a growth chamber calibrated to provide 12 h light/12 h darkness at 28 °C. Three replicates were used for each treatment. The primary radicle and germ lengths were measured after 48 h. The inhibition rate was calculated using the following formula: inhibition rate (%) = ((Lc − Lt)/Lc) × 100%, where Lc is the length of the control and Lt is the duration of the treatment.

*Experiment 2*. Compounds **5**, **6**, and 2,4-D were dissolved in acetone at concentrations of 12.5, 25, 50, 100, 200 μg/mL. The samples were treated in the same way as in *Experiment 1*. The inhibition rates were calculated using the formula presented for *Experiment 1*.

### 4.6. Statistical Analysis

Data were expressed as means ± standard deviation of mean (SD) (*n* = 3). Statistical analysis was performed using Microsoft excel (Microsoft Corp, Redmond, WA, USA) and GraphPad Prism ver. 5 (GraphPad Software, La Jolla, CA, USA). A statistically significant difference was considered when *p* < 0.05.

## 5. Conclusions

Three new Asperugin analogues (**2**–**4**) and a new azaphilone derivative (**5**), together with Asperugin B (**1**) and sydonic acid (**6**), were obtained from an EtOAc extract of *A*. *sparsus* NBERC_28952. The structures were elucidated based on the interpretation of extensive spectroscopic data and single-crystal X-ray diffraction analysis. The herbicidal activities of compounds **1**–**6** were evaluated. Aspersparin D (**5**) and sydonic acid (**6**) exhibited inhibitory activities against the growth of seedlings of *E*. *crusgalli* and *A*. *retroflexus* in a dose-dependent manner, while compound **6** showed an obvious inhibitory effect against seedlings of *A*. *retroflexus* (78.34%) at 200 μg/mL, similar to 2,4-D (80.70%). This is the first report of a chemical investigation of *A*. *sparsus* and on the herbicidal activities of the isolated compounds, which, with further study, might represent new herbicide candidates.

## Figures and Tables

**Figure 1 plants-12-00203-f001:**
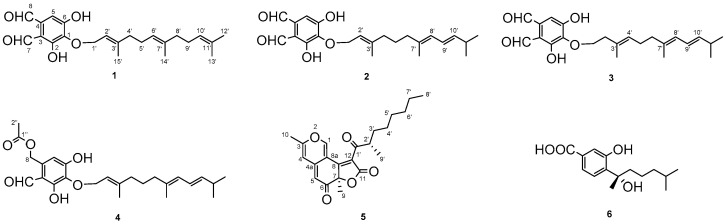
Structures of compounds **1**–**6**.

**Figure 2 plants-12-00203-f002:**
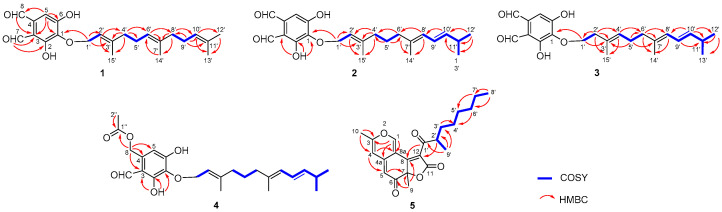
COSY and key HMBC correlations for compounds **1**–**5**.

**Figure 3 plants-12-00203-f003:**
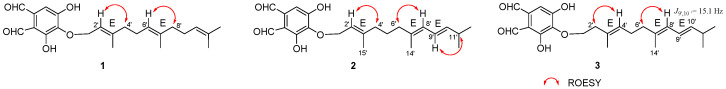
Key NOSEY correlations for compounds **1**–**3**.

**Figure 4 plants-12-00203-f004:**
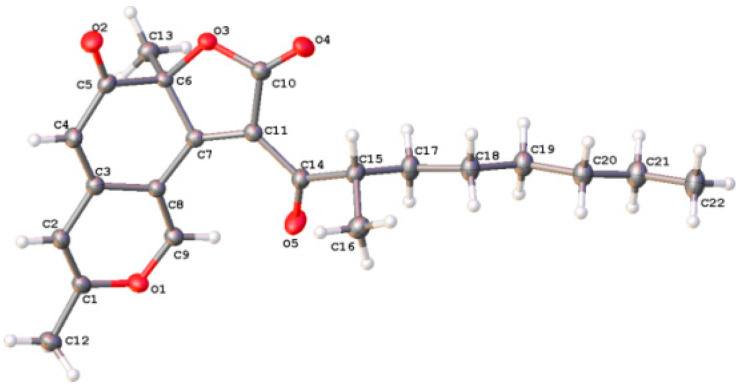
X-ray structure of compound **5**, drawn by Olex2.

**Figure 5 plants-12-00203-f005:**
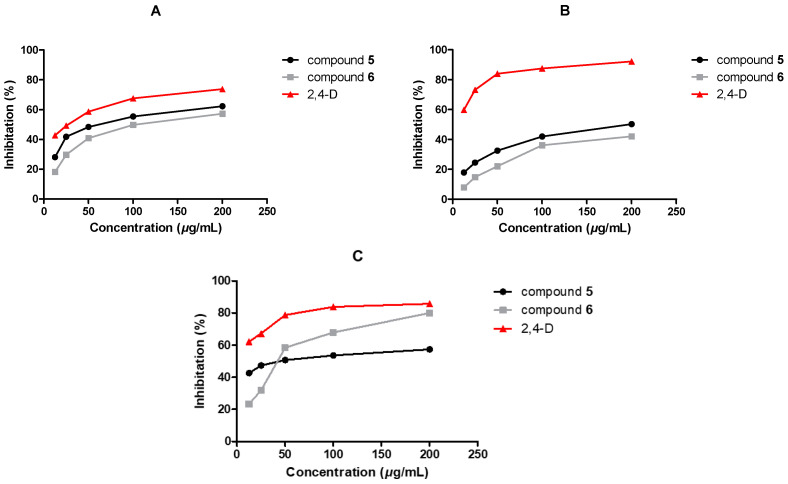
Herbicidal activities of compounds **5**, **6** and 2,4-D (2,4-dichlorophenoxy acetic acid, used as a positive control) on the growth of seedlings of *E*. *crusgalli* and *A*. *retroflexus* at concentrations of 12.5, 25, 50, 100, and 200 μg/mL, respectively. (**A**) shoot of *E*. *crusgalli*, (**B**) root of *E*. *crusgalli*, (**C**) radicle and germ of *A*. *retroflexus*. Values are presented as a percentage of the mean compared to the control (*p* < 0.05).

**Table 1 plants-12-00203-t001:** ^1^H-NMR data for compounds **1**–**4** (500 MHz, Acetone-d_6_, J in Hz).

Position	1	2	3	4
2-OH		12.84 (s)		12.54 (s)
5	7.17 (s)	7.16 (s)	7.18 (s)	6.56 (s)
7	10.73 (s)	10.73 (s)	10.74 (s)	10.12 (s)
8	10.11 (s)	10.09 (s)	10.11 (s)	5.32 (2H, s)
1′	4.78 (2H, d, 7.3)	4.80 (2H, d, 7.3)	4.26 (2H, t, 7.2)	4.68 (2H, d, 7.2)
2′	5.52 (t, 7.3)	5.50 (td, 7.3, 1.2)	2.46 (2H, t, 7.2)	5.50 (m)
4′	2.02 (2H, m)	1.97 (2H, t, 7.4)	5.24 (td, 6.9, 1.1)	1.97 (2H, t, 7.6)
5′	2.05 (2H, m)	1.47 (2H, m)	2.13 (2H, dd, 14.9, 7.3)	1.50 (2H, m)
6′	5.08 (1H, m)	1.88 (2H, t, 7.5)	2.04 (2H, overlapped)	1.92 (2H, t, 7.6)
8′	1.95 (2H, t, 7.2)	5.71 (d, 10.8)	5.77 (d, 10.8)	5.75 (d, 10.7)
9′	2.05 (2H, overlapped)	6.20 (ddd, 15.1, 10.8, 1.0)	6.21 (ddd, 15.1, 10.8, 1.0)	6.22 (ddd, 15.1, 10.8, 1.1)
10′	5.08 (m)	5.53 (dd, 15.2, 7.1)	5.50 (dd, 15.1, 7.0)	5.53 (dd, 15.2, 7.0)
11′	-	2.33 (sext, 6.8)	2.31 (sext, 6.8)	2.33 (sext, 6.8)
12′	1.65 (3H, s)	0.99 (3H, d, 6.8)	0.97 (3H, d, 6.8)	0.99 (3H, d, 6.8)
13′	1.58 (3H, s)	0.99 (3H, d, 6.8)	0.97 (3H, d, 6.8)	0.99 (3H, d, 6.8)
14′	1.58 (3H, s)	1.67 (3H, s)	1.72 (3H, s)	1.70 (3H, s)
15′	1.65 (3H, s)	1.63 (3H, s)	1.67 (3H, s)	1.62 (3H, s)
2″	-	-	-	2.86 (3H, s)

**Table 2 plants-12-00203-t002:** ^13^C-NMR data for compounds **1**–**4** (125 MHz, Acetone-d_6_).

Position	1	2	3	4
1	137.8 s	137.6 s	138.2 s	133.5 s
2	159.7 s	159.7 s	159.2 s	159.4 s
3	113.6 s	113.6 s	113.7 s	113.4 s
4	134.6 s	134.5 s	134.4 s	137.0 s
5	117.6 d	117.6 d	117.6 d	111.3 d
6	157.9 s	157.9 s	157.3 s	158.5 s
7	196.5 d	196.5 d	196.6 d	195.0 d
8	193.1 d	193.0 d	193.1 d	63.2 t
1′	69.5 t	69.4 t	72.2 t	69.2 t
2′	120.8 d	120.8 d	40.4 t	121.2 d
3′	143.4 s	143.6 s	132.3 s	142.8 s
4′	40.5 t	39.9 t	127.5 d	39.9 t
5′	27.5 t	26.5 t	27.3 t	26.6 t
6′	124.7 d	39.8 t	40.7 t	39.9 t
7′	136.0 s	136.6 s	136.4 s	136.7 s
8′	40.4 t	126.2 d	126.2 d	126.2 d
9′	27.1 t	124.8 d	124.7 d	124.8 d
10′	125.2 d	140.1 d	140.1 d	140.0 d
11′	131.8 s	32.1 d	32.1 d	32.2 d
12′	26.0 q	23.0 q	23.0 q	23.0 q
13′	17.9 q	23.0 q	23.0 q	23.0 q
14′	16.2 q	16.5 q	16.6 q	16.5 q
15′	16.5 q	16.3 q	16.3 q	16.3 q
1″	-	-	-	170.6 s
2″				20.9 q

**Table 3 plants-12-00203-t003:** ^1^H (500 MHz) and ^13^C-NMR (125 MHz) data for compounds **5** (CD_3_OD).

Position	δ_H_ (*J* in Hz)	δ_C_
1	8.72 (s)	153.2 d
3	-	160.3 s
4	6.44 (s)	108.1 d
4a	-	146.0 s
5	5.35 (s)	103.7 d
6	-	191.5 s
7	-	87.7 s
8	-	165.2 s
8a	-	111.1 s
9	1.69 (3H, 3)	24.6 q
10	2.28 (3H, s)	17.9 q
11	-	168.3 s
12	-	124.2 s
1′	-	200.7 s
2′	3.58 (m)	43.3 d
3′	1.56 (m), 1.34 (m)	33.3 t
4′	1.32 (2H, m)	28.9 t
5′	1.04 (2H, m)	27.5 t
6′	1.35 (2H, m)	26.3 d
7′	1.20 (2H, m)	22.1 s
8′	0.86 (t, 7.3)	13.0 t
9′	1.16 (3H, d, 6.6)	13.8 t

**Table 4 plants-12-00203-t004:** Inhibitory effects of compounds **1**−**6** on the growth of the seedlings of *E*. *crusgalli* and *A*. *retroflexus*.

Compounds *^a^*	Inhibition Rates (%) *^b^*		
	*E*. *crusgalli*		*A*. *retroflexus*
	Root	Shoot	Radicle and Germ
**1**	16.15 ± 0.97	13.93 ± 0.53	22.25 ± 1.15
**2**	23.15 ± 1.19	14.92 ± 0.29	18.42 ± 0.90
**3**	23.22 ± 1.13	16.11 ± 0.38	15.31 ± 1.38
**4**	30.05 ± 1.50	21.61 ± 1.06	38.01 ± 1.56
**5**	51.74 ± 0.60	56.66 ± 0.67	53.38 ± 0.52
**6**	46.78 ± 0.63	52.74 ± 0.82	78.34 ± 1.39
**2,4-D** * ^c^ *	94.18 ± 0.49	77.84 ± 0.45	80.70 ± 0.78

*^a^* All the compounds were tested at a concentration of 200 μg/mL. *^b^* Values are presented as a percentage of the mean compared to the control (mean ± SD). *^c^* 2,4-dichlorophenoxy acetic acid, used as a positive control.

## Data Availability

Not applicable.

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
