# Peer review of "Secondary Metabolites from Aspergillus sparsus NBERC_28952 and Their Herbicidal Activities"

_plants, 2023, doi:10.3390/plants12010203_

Round 1
Reviewer 1 Report
Manuscript: Plants-2094628
A good deal of attention has been paid to the organization of the manuscript and the application of the research appears interesting. However, it may be necessary to make some modifications to the research in order to improve its scientific quality.
-In the Introduction, the field of interest in which the research is focused should be highlighted more, emphasizing the current conditions of the crops and their problems.
-The extraction method needs to be described in more detail in paragraph "3.3. Cultivation, Extraction, and Isolation" by referencing its protocol.
-Statistical analysis should be conducted on the results relating to the inhibitory effects and herbicidal activities (Tables 4 and 5). I would appreciate it if you would add a paragraph in the Materials and Methods section to describe the statistical analysis that was performed and to report the significant differences in Table 4 and Figure 5.
-A discussion paragraph should be included due to the high impact of the research and its potential application implications. The discussion paragraph should address how the research will improve current crop issues and how further research will be directed in order to solidify and consolidate these findings.
Reviewer 2 Report
The article is very interesting, but it needs improvements. The Introduction part is insufficiently developed. Information from the specialized literature must be added, including new bibliographic titles to the References.
In the Material and Method it is not specified when was conducted the experiment and to whom the calculation methods belong (for exemple inhibition rate. Anyway the method of determining the herbicide effect is not clear.
In the Results part, all abbreviations in tables and figures must be explained in footnotes.
I think the herbicidal activities of the secondary metabolites from Aspergillus Sparsus is not studied sufficiently (in the conditions where the title of the article suggest this).
Round 2
Reviewer 1 Report
Most of the requested changes were made by the authors, improving the quality of the work. However, a few requested changes were not adequately addressed, such as:
As mentioned in the response report, the authors report: "Moreover, a detailed description of Cultivation, together with spectra data of compound 6, were also added to paragraph 3.3.". I would like to know the line in the text where this can be found.
-The discussion paragraph is too short, as in order to provide a more convincing evaluation of the scientific impact of the work, it should critically analyze the obtained results by placing them in the context of the research in which the study was conducted and describe their relationship to the scientific state of the art.
Reviewer 2 Report
The paper can be published in present form.
Author Response
The reviewer suggested that the paper can be published in present form.